# Length of Washout Period After Remission Does Not Influence Relapse Risk in Patients with Acute Myeloid Leukemia Treated with Hypomethylating Agents Combined with Venetoclax

**DOI:** 10.3390/jcm14145007

**Published:** 2025-07-15

**Authors:** Fanny Erika Palumbo, Andrea Duminuco, Laura Longo, Daniela Cristina Vitale, Cinzia Maugeri, Serena Brancati, Marina Silvia Parisi, Giuseppe Alberto Palumbo, Giovanni Luca Romano, Filippo Drago, Francesco Di Raimondo, Lucia Gozzo, Calogero Vetro

**Affiliations:** 1Division of Hematology, A.O.U. Policlinico “G.Rodolico—S. Marco”, 95123 Catania, Italy; fannypalumbo@gmail.com (F.E.P.); maugericinzia@hotmail.com (C.M.); marinaparisi@hotmail.it (M.S.P.); palumbo.ga@gmail.com (G.A.P.); francesco.diraimondo@unict.it (F.D.R.); 2Clinical Pharmacology Program/Regional Pharmacovigilance Centre, A.O.U. Policlinico “G.Rodolico—S. Marco”, 95123 Catania, Italy; longo@policlinico.unict.it (L.L.); danielac.vitale@gmail.com (D.C.V.); serena.brancati@gmail.com (S.B.); f.drago@unict.it (F.D.); luciagozzo86@icloud.com (L.G.); 3Department of Scienze Mediche Chirurgiche e Tecnologie Avanzate “G.F. Ingrassia”, University of Catania, 95123 Catania, Italy; 4Faculty of Medicine and Surgery, University of Enna “Kore”, 94100 Enna, Italy; giovanniluca.romano@unict.it; 5Hematology and Bone Marrow Transplantation Unit, Hospital of Bolzano (SABES-Azienda Sanitaria dell’Alto Adige), Teaching Hospital of Paracelsus Medical University, 39100 Bolzano, Italy; gerovetro@gmail.com

**Keywords:** acute myeloid leukemia, venetoclax, cytopenia management, washout period

## Abstract

**Background/Objectives:** The combination of venetoclax (VEN) and hypomethylating agents (HMA), such as azacitidine (AZA) or decitabine (DEC), has transformed the treatment landscape for acute myeloid leukemia (AML) in patients unfit for intensive chemotherapy. However, optimal management of neutropenia and the impact of post-remission treatment interruptions (washouts) remain unclear. This study aimed to evaluate the safety and efficacy of post-remission washouts and their effect on clinical outcomes. **Methods:** We conducted a retrospective single-center study of 44 AML patients treated with HMA/VEN between 2020 and 2021. Clinical, molecular, and treatment-related data were collected, including treatment duration, post-remission washout duration, response rates, disease-free survival (DFS), and overall survival (OS). Statistical analyses included Fisher’s exact test and univariate and multivariate Cox models. **Results:** Overall, 61% of patients responded to therapy, with significantly higher response rates among those potentially eligible for the VIALE-A trial (86% vs. 39%, *p* = 0.002). Neither treatment duration nor post-remission washout length was associated with DFS or OS. DFS was significantly longer in patients treated with AZA compared to DEC (*p* = 0.006). Median OS was 7.7 months, with longer OS observed in patients who met VIALE-A trial eligibility criteria (*p* = 0.021). Achieving complete remission (CR) was associated with improved OS (14.5 months). **Conclusions:** Post-remission treatment interruptions (washouts) did not negatively impact DFS or OS, suggesting they may be a safe strategy to support hematologic recovery. However, the choice of HMA appears to influence response duration, with AZA outperforming DEC in maintaining disease control.

## 1. Introduction

The introduction of venetoclax (VEN) in combination with hypomethylating agents (HMAs) such as azacitidine (AZA) and decitabine (DEC) has revolutionized the treatment landscape for unfit patients with acute myeloid leukemia (AML) [1,2]. The VIALE-A trial evaluated the efficacy and safety of the combination therapy venetoclax plus AZA compared to AZA alone in AML patients ineligible for intensive chemotherapy. The study demonstrated a statistically significant improvement in overall survival (OS) for patients receiving the combination therapy, with a median OS of 14.7 months compared to 9.6 months in the control group (hazard ratio [HR] for mortality: 0.66; *p* < 0.001) [3]. The combination therapy also led to a higher rate of complete remission (CR) (36.7% vs. 17.9%; *p* < 0.001) and a greater proportion of patients achieving CR or CR with incomplete hematologic recovery (CRi) (66.4% vs. 28.3%; *p* < 0.001). However, treatment-associated adverse events were more frequent in the combination therapy group. Febrile neutropenia occurred in 42% of patients receiving venetoclax, compared to 19% in the control group, and infectious complications were reported in 84% vs. 67%, respectively.

The duration of neutropenia following treatment with venetoclax varies, typically ranging from a few weeks to several months. Studies have shown that the median duration of grade 3 neutropenia can be around 28 days, and some patients may experience prolonged neutropenia for up to 46 days, with a higher risk of related death [4,5]. The duration of neutropenia appears not to be affected by concomitant treatment with CYP3A4 inhibitors (mostly antifungals), in contrast to platelet recovery, which seems to be influenced by concomitant CYP3A4 inhibitor treatment [6]. Some schedules also allow a break of up to 10 weeks if a patient is already in CR [7].

Delaying cycles is a good strategy to minimize the risk of febrile neutropenia occurrence and is also the strategy reported in the drug datasheet [8].

The VIALE-A trial established that modifications in VEN dosing due to neutropenia should not be implemented until complete remission is achieved, and neutropenia management should be guided by bone marrow assessment rather than automatic dose reductions [3]. Until remission is achieved, supportive care measures, including prophylactic antimicrobials and blood product transfusions, should be prioritized over altering treatment protocols. After achieving remission, treatment interruptions may occur if the blast count drops below 5%, with therapy resuming once the absolute neutrophil count (ANC) surpasses 500 cells/µL or is stimulated using G-CSF (filgrastim). In cases of grade 4 neutropenia post-remission, VEN treatment should be delayed until the ANC recovers to exceed 500 cells/µL. Persistent or recurrent grade 4 neutropenia warrants a bone marrow biopsy to reassess disease status. If neutropenia recurs after remission, subsequent VEN cycles should be shortened by 7 days (e.g., 21-day cycles instead of 28 days) [9]. The VIALE-A trial also modified treatment for patients experiencing both neutropenia and thrombocytopenia, delaying VEN/AZA therapy until ANC recovery and platelet count exceeded 50 × 10^3^/µL [3].

Apart from safety concerns, some questions may arise regarding the duration of the therapy break after remission. Specifically, the question is whether a treatment washout period should be a safe and viable strategy to support hematologic recovery without compromising outcomes.

Trying to address this question, we analyzed data from our institution regarding the length of post-remission washout and the risk of disease recurrence.

## 2. Materials and Methods

This prospective observational study began in March 2020 at the University Hospital of Catania and enrolled newly diagnosed AML patients, who were either ineligible for intensive induction chemotherapy or aged 75 years or older, and were treated with a combination of hypomethylating agents (HMAs) and venetoclax (VEN). The study was part of a broader pharmacovigilance program conducted at our institution to investigate the safety of off-label treatment strategies. AML patients included in this pharmacovigilance sub-study met the eligibility criteria established by Italian Law 648/96. Preliminary safety data have already been published [1], primarily focusing on the occurrence of adverse events. In the present post-hoc analysis, we extended the follow-up period. We expanded the sample size, with the primary objective of evaluating the impact of washout duration during cytopenia management on patient safety. AZA was administered subcutaneously at 75 mg/m^2^ of body surface area (BSA) from days 1 to 7, while DEC was given intravenously at 20 mg/m^2^ on days 1 to 5. VEN was taken orally once daily with food, starting the same day as the hypomethylating agent (HMA). The dosing followed an initial ramp-up phase, with 100 mg on day 1, 200 mg on day 2, and the target dose of 400 mg reached by day 3, continuing until the end of the cycle. In subsequent cycles, VEN was consistently administered at 400 mg daily. Each cycle was planned for 28 days, unless unacceptable toxicity, e.g., profound neutropenia (i.e., neutrophils lower than 500/μL) occurred.

To manage cytopenia and its clinical effects, VEN treatment was interrupted as needed for blood count recovery. Treatment interruption criteria were defined as follows: Venetoclax was withheld in case of marrow remission in the event of persistent grade 4 neutropenia (ANC < 500/μL) or thrombocytopenia (platelets < 25,000/μL). Therapy was resumed only after adequate hematologic recovery (ANC ≥ 500/μL and platelets ≥ 25,000/μL), and venetoclax was reintroduced for further cycles, with the option to hold it in case of newly occurring grade 4 cytopenias. Additionally, dose adjustments were made to account for the effects of prophylactic anti-infective agents on VEN pharmacokinetic properties. Notably, antifungal prophylaxis (AP) was performed until hematological recovery in responding patients. Dose adjustments of VEN were applied according to the type of CYP3A4 inhibitor used: a 75% dose reduction was implemented in patients receiving strong CYP3A4 inhibitors (e.g., posaconazole or voriconazole), and a 50% reduction was applied for those treated with moderate inhibitors such as fluconazole. No dose adjustment was required for patients receiving caspofungin. Figure 1 provides an overview of the intended treatment pathway.

Response to treatment was assessed according to the 2017 European Leukemia Net (ELN) criteria. CR was defined by the presence of an absolute neutrophil count (ANC) greater than 1000/μL, a platelet count exceeding 100,000/μL, independence from red blood cell transfusions, and less than 5% bone marrow blasts. In cases of complete remission with partial hematologic recovery (CRh), patients were required to have bone marrow blasts below 5% and no evidence of residual disease, along with partial recovery of peripheral blood counts, specifically an ANC value above 500/μL and platelet counts over 50,000/μL. Morphologic leukemia-free state (MLFS) was determined by the presence of less than 5% blasts in the bone marrow, the absence of Auer rods, and no detectable extramedullary disease; hematologic recovery was not required for this category. Partial response (PR) was defined as a reduction in bone marrow blast percentage to between 5% and 25%, with at least a 50% decrease from baseline values. Stable disease (SD) included patients who did not meet criteria for CR, CRh, PR, or MLFS, and who did not show signs of progression. Progressive disease (PD) was characterized by an increase in bone marrow blast percentage and/or rising absolute blast counts in the peripheral blood. Overall response rate (ORR) encompassed patients in CR or CRh, or MLFS. Treatment failure encompassed refractory patients and also patients who died before disease re-evaluation.

For each responding patient (i.e., patients in CR/CRh/MLFS) and for every treatment cycle administered, the number of days during which venetoclax was effectively administered (treatment days) and the number of days between VEN stoppage and the start of the next cycle (washout days) were recorded. For each responding patient, the average number of treatment and washout days across all received cycles was calculated.

### 2.1. Diagnostic Work-Up and Ancillary Assessment

Before initiating treatment, patients underwent comprehensive blood testing, including a complete blood count as well as assessments of liver and renal function. A bone marrow aspirate and/or biopsy was performed locally for morphological, histological, flow cytometric, and cytogenetic analysis. Additionally, a cardiological evaluation, including echocardiography, was performed, along with a pneumonological assessment that included pulmonary function tests, when deemed necessary.

Q-banding cytogenetic analysis was conducted locally for all patients, with aberrations characterized according to the Medical Research Council Criteria [10]. Molecular testing was performed according to local protocols, including assessing *NPM1* and *FLT3*-ITD mutations in all patients [11].

Fitness evaluation was based on SIE/SIES/GITMO criteria [12], and AML comorbidity index (AML-CI) was calculated, including the following parameters: LDH; cardiac dysfunction; liver dysfunction; infection; peptic ulcer; heart valve disease; albumin level < 3.5 g/dL; platelet count < 20 × 10^3^/uL [13]. AML-CI was considered low for scores of 0–1, intermediate for scores of 2–3, and high for scores ≥ 4. For stratifying patients, we referred to the inclusion criteria of the VIALE-A trial, available in the Appendix A.

Neutropenic fever (NF) was defined as the presence of fever (oral temperature ≥ 38.3 °C in a single measurement or remains at ≥38.0 °C for at least one hour) in a patient with neutropenia, i.e., absolute neutrophil count (ANC) below 500 cells/μL.

### 2.2. Statistical Analysis

Dichotomous variables were compared using chi-square or Fisher’s exact test, as appropriate. Continuous variables were compared using Student’s *t*-test or the Wilcoxon rank sum test when normal distribution was not confirmed. Disease-free survival (DFS) was defined as the time from the date of CR to the occurrence of relapse or death from any cause, whichever came first. OS was calculated from the first day of treatment to the date of death from any cause or the last follow-up date. Survival curves were built using the Kaplan–Meier method, and univariate survival analysis was performed using the Log-rank test. A Cox proportional hazard model was built for each multivariate survival analysis, including only the variables that respected the proportional risk assumption. A multivariate logistic regression model was constructed to evaluate the risk of death, relapse, and DFS, incorporating only variables with a *p*-value < 0.05 from the initial univariate analysis. Statistical analyses were performed using IBM SPSS v22© (IBM, Armonk, NY, USA).

Adverse events (AEs) during treatment were defined and graded according to the Common Terminology Criteria for Adverse Events (CTCAE) version 5.

## 3. Results

### 3.1. Patient Features

Forty-four patients were enrolled between 2020 and 2021. The median age was 72, ranging from 47 to 86 years. Twenty-eight out of forty-four patients were male (63%).

All of them were treated with HMA/VEN as first-line cause not deemed fit for intensive chemotherapy according to SIE/SIES/GITMO criteria [12].

All but 21 patients would have met the eligible criteria for the VIALE-A trial (44.7%), as they met the inclusion criteria. Conversely, 23 patients (52.3%) would not have met the eligibility criteria due to the following reasons: five (21.7%) had been previously treated with AZA for myelodysplastic syndrome (MDS), seven (30.4%) had an ECOG Performance Status (PS) of 4, seven (30.4%) had a PS of 3 and were 75 years or older, one (4.3%) had a history of systemic mastocytosis, and three (13.2%) had a prior myeloproliferative disease (two of whom had already received treatment with ruxolitinib).

The AML-CI was 4 in seven patients (15.9%), 3 in twelve patients (27.3%), 2 in eleven patients (25%), and 0–1 in fourteen patients (31.8%).

The baseline median hemoglobin (Hb) level was 8.04 g/dL (normal range 13.8–17.2 for men and 12.1–15.1 for women); median white blood cell (WBC) was 1865/μL, ranged between 680/μL and 210,000/μL (normal value 4000/μL–11,000/μL); and median platelets were 34,000/μL, ranged between 2000/μL and 130,000/μL (normal value 150,000/μL–450,000/μL).

Of the 44 patients, 6 (13%) had an *NPM1* mutation and 3 (6%) had *FLT3*-ITD, with 1 of them also having a concomitant *NPM1* mutation. *IDH1* and *IDH2* were evaluated in 30 patients and were found to be positive in 4 (9%) and 3 (6%) patients, respectively. Complete molecular analysis was available for 19 patients, revealing the following mutations: *TP53* in 1 patient (2%); *ASXL1* in 3 patients (6%); *TET2* in 3 patients (6%); *CEBPA* in 1 patient (2%); *MPL* in 1 patient (2%); *NRAS* in 1 patient (2%); *PTPN11* in 1 patient (2%); *RB1* in 1 patient (2%); *RUNX1* in 2 patients (4%); *SF3B1* in 2 patients (4%); *STAG2* in 2 patients (4%); *DNMT3A* in 2 patients (4%); and *NF1* in 1 patient (2%). Karyotype analysis was evaluable in 41 patients (93.2%), while it failed in 3 cases. A total of 7 patients (15.9%) had high-risk cytogenetics, and 3 of these (42.9%) had a complex karyotype.

A total of 13 patients (30%) underwent combination therapy with DEC/VEN and 31 (70%) with AZA/VEN.

A total of 21 patients (48%) received AP. The agents administered included posaconazole in 5 patients (11%), voriconazole in 1 patient (2%), fluconazole in 10 patients (23%), and caspofungin in 5 patients (11%). Neutropenic fever (NF) was observed in 6 out of 23 patients (26.1%) who did not receive antifungal prophylaxis (AP), compared to 12 out of 21 patients (57.1%) who did (*p* = 0.065). Among patients receiving AP, the incidence of NF was 45.5% (5/11) in those treated with a moderate CYP3A4 inhibitor, 60% (3/5) in those receiving a strong CYP3A4 inhibitor, and 80% (4/5) in those receiving caspofungin.

### 3.2. Treatment Outcome

A total of 38 pts (86%) were assessed for treatment response at the end of the first cycle. CR was reached in 3 out of 38 pts (8%), CRh in 8 (21%), MLFS in 10 (26%), PR in 5 (13%), and SD in 12 (32%). All patients in PR and 1 patient with SD reached CR after the second cycle. Among the 27 responders (including CR/CRh/MLFS), 7 (26%) relapsed and 5 (18.5%) died due to NF (20%). Those pursuing treatment received a median of five cycles (range 2–16).

Regarding ORR, a statistically significant relationship was observed comparing patients that would have been excluded from the VIALE-A trial and patients that would have been included (*p* = 0.002), with higher CR rates among patients who would have been included (85.7%) compared to those who would not have been included (39.1%). Other variables, including gender, treatment type (AZA vs. DEC), AP, molecular mutations (*NPM1*, *FLT3 ITD*, *IDH1*, *IDH2*), cytogenetic risk, AML-CI, and complex karyotype, did not show statistically significant associations with treatment response (Table 1).

Regarding the DFS analysis, the treatment group was the only factor significantly associated. Patients receiving AZA exhibited a substantially longer DFS compared to those treated with DEC (median DFS: not reached vs. 7.9 months, *p* = 0.006). Other variables, including gender, possible eligibility for the VIALE-A trial, baseline AML-CI, *NPM1* mutation, and cytogenetics, did not show statistically significant differences in DFS distributions. Univariate Cox regression analysis revealed that, among the tested variables, treatment type and WBC count were significantly associated with DFS (Table 2). Patients treated with DEC had a significantly higher risk of disease recurrence or progression compared to those receiving AZA, with a hazard ratio (HR) of 11.3 (95% CI: 1.34–95.75; *p* = 0.026). Similarly, WBC count was associated with worse DFS (HR = 1.000; *p* = 0.015). In the multivariate analysis including both variables, treatment with DEC remained an independent predictor of inferior DFS (HR = 16.37, 95% CI: 1.30–207.05; *p* = 0.031), while WBC count lost statistical significance (HR = 1.000; 95% CI: 1.000–1.001; *p* = 0.288). It should be noted that average treatment and washout days, i.e., the number of days during which the patients received or did not receive VEN, calculated from the various cycles done by patients, were 20 ± 4 and 20 ± 25, respectively, and did not differ based on HMA; moreover, these variables did not relate with DFS as well (HR 0.89; 95%CI 0.7–1.1; HR 0.98; 95%CI 0.9–1.1, respectively). Conversely, the washout period varied according to the type of AP received. Patients who did not receive any AP (n = 11) had a mean washout duration of 26.0 days (±35.0 SD). Among those receiving moderate CYP3A4 inhibitors (n = 7), the mean washout period was 10.5 days (±10.5 SD), while for patients treated with strong CYP3A4 inhibitors (n = 4), the mean was 10.7 days (±2.9 SD). Patients receiving caspofungin (n = 5) had a mean washout duration of 27.7 days (±8.4 SD). Although the washout duration did not differ significantly between patients not receiving AP and those receiving caspofungin, it was significantly shorter in patients treated with moderate or strong CYP3A4 inhibitors compared to those receiving caspofungin (*p* = 0.02 and *p* = 0.009, respectively).

After a median follow-up of 11.4 months (range 0.5–22.2), median OS was 7.7 months (95% CI: 5.9–9.4). Stratified analyses were performed to assess survival differences across various clinical and biological variables (Table 3). Notably, patients who would have been eligible for the VIALE-A trial had significantly longer OS compared to those not eligible, with median OS estimates of 8.4 months versus 5.8 months, *p* = 0.021, respectively (Figure 2). Conversely, survival did not differ significantly by gender, AP (shown in Figure 3), *NPM1* mutation status, or treatment regimen (AZA vs. DEC). Notably, a landmark OS analysis performed at the time of disease re-evaluation showed that responding patients (accounting for 27 patients) had a median OS of 14.5 months (95% CI: 5.2–23.8), which was significantly longer than that observed in refractory patients (accounting for 11 patients), whose median OS was 4.9 months (95% CI: 1.9–8.0; *p* < 0.001). Cox regression models were employed to assess the impact of individual covariates on OS (Table 4). In univariable analysis, potential exclusion from the VIALE-A trial emerged as a significant predictor of lower survival (HR = 2.50, 95% CI: 1.12–5.61; *p* = 0.026). Other variables, including gender, treatment type (AZA vs. DEC), AP, AML-CI, and *NPM1* mutation, were not statistically significant predictors. Importantly, neither washout duration (HR = 0.98, 95% CI: 0.92–1.05, *p* = 0.611) nor overall treatment duration (HR = 0.91, 95% CI: 0.77–1.08, *p* = 0.276) was significantly associated with OS. These findings indicate that timing variables, including time off therapy or cumulative exposure, do not independently influence prognosis in this cohort.

## 4. Discussion

The introduction of VEN in combination with HMAs has significantly altered the treatment landscape for AML patients ineligible for intensive chemotherapy, as demonstrated in the VIALE-A trial and supported by multiple real-world experiences [3]. The drug was approved in Europe for the treatment of treatment-naïve AML patients in 2021 [13] and in Italy for reimbursement in 2023 [14]. However, according to Italian Law 648/96, the Italian Medicines Agency has allowed early access to the drug since 2020 [14,15], monitoring prescriptions through a registry [16].

In our cohort, the median OS was 7.7 months, lower than the 14.7 months reported in the VIALE-A trial but aligned with outcomes observed in routine clinical practice [3,17]. Indeed, patients from our cohort who would have been excluded from the VIALE-A trial, due to factors such as prior MDS or poor PS, demonstrated lower OS than those who met eligibility criteria. This observation may reflect underlying differences in disease biology influencing the outcome. However, our data are not in line with a recent study (namely “VALOR”), which indicated no differences in the outcome depending on the potential eligibility for the VIALE-A trial. Differences could theoretically rely on the baseline patient features, in particular, the VALOR study had no patients with a PS of 4; however, patients already treated with HMA were included [18]. More data coming from real-world analyses will further clarify these aspects, for example, the study AML2320 from the Gruppo Italiano Malattie Ematologiche Neoplastiche dell’Adulto (GIMEMA) [17]. Regarding PS, it is noteworthy that in Italy from 2023, the use of VEN in treatment-naïve patients has been limited to patients with a PS 0–2, or 3 if <75 years old [15], while from 2020 up to 2023 (our study period), PS was not a restriction factor for treatment [19]. Regarding this aspect, although it is comprehensible that patients with impaired PS would have a lower benefit from the survival point of view (although a comparison with best supportive care or HMA alone has not been performed, yet), it is not foreseen to think that these patients would have had an amelioration of the quality of life [20]. Unfortunately, our study was not designed to investigate this aspect, which deserves more attention in the future. Moreover, the Italian Rete Ematologica Lombarda (REL) group analyzed the impact of HMA + AZA in frail patients, reporting a median overall survival (OS) of 2.9 months in treated patients, which was significantly longer compared to 1.1 months in those receiving best supportive care alone (*p* = 0.04), but the clinical impact of this treatment in these patients should be further investigated [21].

As expected, CR emerged as the strongest predictor of OS in our cohort, with a median OS of 15.2 months among responders, confirming the central role of early disease control. This is consistent with the VIALE-A trial, where early responders, defined as patients achieving CR or CRi by the end of cycle 2, reached a median OS of 24.4 months, and MRD-negative responders had an OS of 34.2 months compared to 18.7 months in MRD-positive patients [22,23].

A major focus of our analysis was the impact of post-remission treatment interruptions, particularly washout periods. Neither the length of these delays nor the total time off treatment had a negative impact on OS or DFS, provided remission was achieved, also in cases of very long washout, i.e., more than 40 days. These results are in line with a post hoc analysis of the VIALE-A and VIALE-C trials, which showed that post-remission G-CSF use and treatment delays were not associated with inferior survival [23]. Cycle flexibility was a prominent feature in both our cohort and the VIALE-A trial. A post hoc analysis of the VIALE-A trial evaluated VEN dosing modifications in patients with AML who achieved CR [24]. Among these responders, 69% underwent at least one post-remission cycle with a reduced VEN duration (≤21 days). The median time from achieving CR/CRh to the first shortened cycle was 92 days. Importantly, OS was not compromised by these modifications: patients who converted to 21/28-day dosing schedules, either early or late, demonstrated favorable 2-year OS estimates of 70.1% and 65.0%, respectively. These findings support the clinical feasibility and safety of dose-adjusted maintenance strategies in this population. These findings are further supported by other recent studies, which report no correlation between VEN duration (≤14 days vs. >14 days) and hematologic toxicity, achieving similar CR rates but a lower rate of AEs with shorter VEN schedules [25,26,27]. In a separate real-world analysis, AML patients achieving remission for ≥12 months on VEN-based therapy had a median treatment-free remission (TFR) of 45.8 months after elective discontinuation, highlighting the possibility of sustained remission off treatment [28]. Similarly, data from the French Innovative Leukemia Organization (FILO) Group showed that patients who discontinued VEN and/or AZA for reasons other than progression experienced a median TFR of 15 months and median OS of 44 months [29]. These results suggest that discontinuation in selected patients may be a viable strategy, particularly in those with durable responses to treatment. A multicenter community study also demonstrated that VEN schedule modifications after remission, such as shortened duration or cycle delays, were not associated with worse outcomes among responders [30]. These findings support a shift away from rigid 28-day cycles toward a patient-tailored, response-adapted approach, particularly after remission. Structured dose holds or delays, used to manage prolonged cytopenia, infections, or logistical challenges, can be safely implemented without compromising efficacy. On the other hand, we found a significant difference in washout duration across AP groups. This likely reflects differences in clinical recovery rather than treatment strategy. In our cohort, washout was not dictated by clinicians but rather depended on the patient’s ability to recover from prior toxicity or infection. Patients receiving azole-based prophylaxis (moderate or strong CYP3A4 inhibitors) exhibited shorter washout periods, suggesting faster hematologic or clinical recovery, and potentially a more favorable clinical status. Conversely, patients receiving caspofungin, or no prophylaxis at all, may have required longer recovery periods due to more severe infectious complications or delayed hematologic reconstitution. In this context, the delay in treatment due to the need for hematologic recovery may have theoretically conferred a survival advantage, as the cytopenic patient was not subjected to the burden of a subsequent chemotherapy cycle during a particularly vulnerable phase. This could partly explain why the duration of the washout period did not correlate with OS or DFS, as it may reflect a protective delay rather than a marker of treatment interruption or disease aggressiveness.

However, limitations of our study include its retrospective, single-center design, small sample size, absence of MRD monitoring, and heterogeneity in molecular profiling, which restricts broad generalizability. We acknowledge that our study was conducted on a convenience sample [31], consisting of all consecutive patients treated at our institution who met predefined clinical and data completeness criteria. Although not randomized, this sampling strategy enabled the inclusion of a real-world, unselected population. Importantly, this approach reflects routine clinical practice and enhances the internal consistency of the analysis. The lack of statistically significant correlations should thus be interpreted in light of the exploratory nature of the analysis, and not as definitive evidence of the absence of association. Larger, multicenter studies are warranted to explore these hypotheses further.

Nonetheless, our results add to the growing body of evidence supporting individualized, remission-adapted scheduling of HMA/VEN therapy in unfit AML patients. Future prospective studies should validate these findings with integrated MRD assessment and dynamic cytopenia monitoring to optimize treatment strategies.

## 5. Conclusions

Our real-world data, together with emerging evidence from recent studies, support a paradigm shift in the management of AML patients treated with HMA/VEN combinations. Achieving remission remains the key determinant of survival, while fixed 28-day VEN cycles appear unnecessary in the post-remission setting. Flexible, remission-adapted dosing strategies—including temporary treatment interruptions or schedule reductions—can be safely adopted without compromising efficacy. These approaches not only optimize tolerability and outpatient feasibility but may also sustain long-term outcomes in selected patients. Future prospective trials incorporating MRD and cytopenia-guided treatment modulation are essential to refine this individualized framework and further improve the therapeutic index of VEN-based regimens in unfit AML patients.

## Figures and Tables

**Figure 1 jcm-14-05007-f001:**
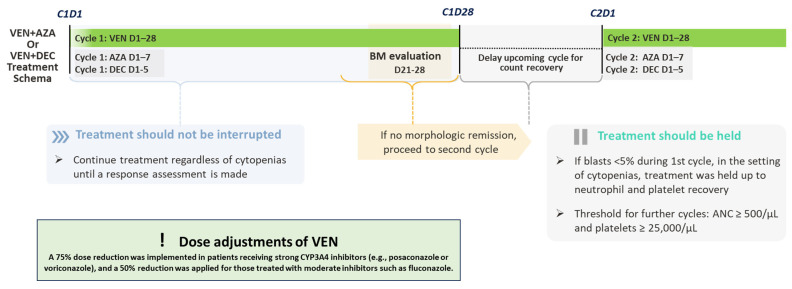
Treatment workflow for studied patients. The figure illustrates the therapeutic course for patients treated with HMA (AZA: azacytidine or DEC: decitabine) plus venetoclax (VEN). Cycle 1 includes VEN from day 1 to 28, with either azacitidine (days 1–7) or decitabine (days 1–5). Bone marrow (BM) evaluation is performed between days 21 and 28. If no morphologic remission is achieved, treatment proceeds to cycle 2. In the setting of cytopenias, treatment continuation or delay is guided by hematologic recovery, with a threshold of absolute neutrophil count (ANC) ≥ 500/μL and platelets ≥ 25,000/μL. Venetoclax dose was reduced by 75% in patients receiving strong CYP3A4 inhibitors and by 50% in those treated with moderate inhibitors.

**Figure 2 jcm-14-05007-f002:**
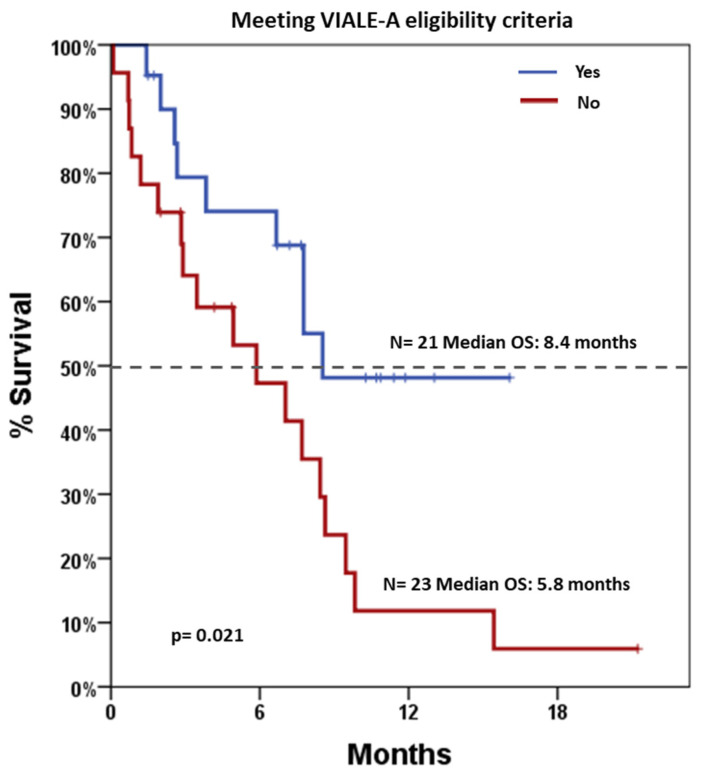
Overall Survival based on patients’ potential eligibility for the VIALE-A trial. Blue line indicates patients meeting the eligibility criteria (21 patients), and red line indicates patients not meeting the eligibility criteria (23 patients). Dotted line indicates 50% survival.

**Figure 3 jcm-14-05007-f003:**
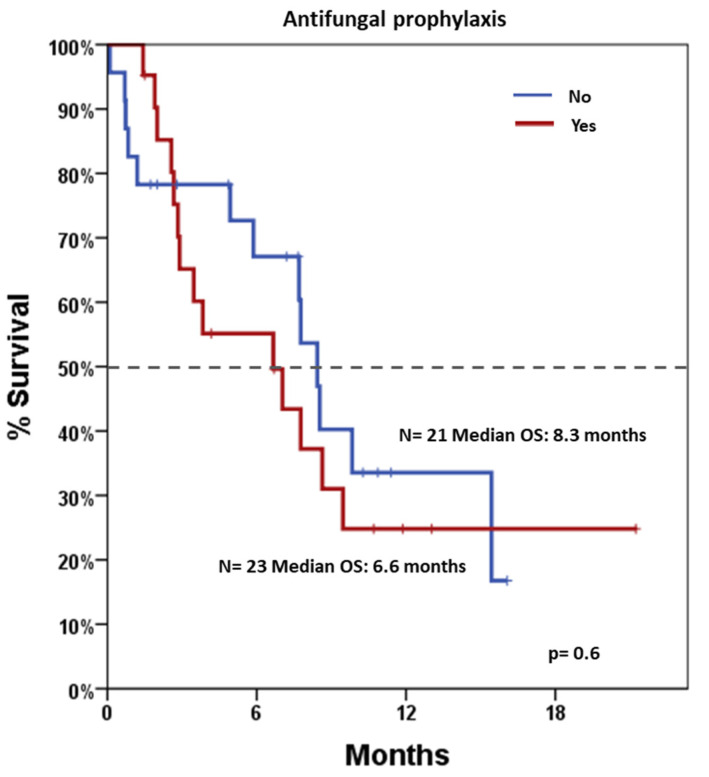
Overall survival according to antifungal prophylaxis. Blue line indicates patients undergoing and red line indicates patients not undergoing antifungal prophylaxis. Dotted line indicates 50% survival.

**Table 1 jcm-14-05007-t001:** Remission rates according to baseline features.

Parameter	Category	ORR		Failure		Total	*p*-Value
		n	%	n	%		
Gender	Female	10	62.5	6	37.5	16	0.907
	Male	17	60.7	11	39.3	28	
**Eligible for VIALE-A trial**	**Yes**	**18**	**85.7**	**3**	**14.3**	**21**	**0.002**
	**No**	**9**	**39.1**	**14**	**60.9**	**23**	
Treatment	AZA	17	54.8	14	45.2	31	0.170
	DEC	10	76.9	3	23.1	13	
Antifungal prophylaxis	No	11	47.8	12	52.2	23	0.054
	Yes	16	76.2	5	23.8	21	
AML-CI	0–1	7	50	7	50	14	0.547
	2–3	15	65.2	8	34.8	23	
	4	5	71.4	2	28.6	7	
*NPM1*	Mutated	4	66.7	2	33.3	6	0.738
	Wild type	22	59.5	15	40.5	37	
*FLT3*-ITD	Mutated	2	66.7	1	33.3	3	0.706
	Wild type	24	60.0	16	40.0	40	
*IDH1*	Mutated	3	75.0	1	25.0	4	0.667
	Wild type	16	64.0	9	36.0	25	
*IDH2*	Mutated	2	66.7	1	33.3	3	1.000
	Wild type	18	66.7	9	33.3	27	
Cytogenetic Risk	Intermediate	22	66.7	11	33.3	33	0.4
	Adverse	3	37.5	5	62.5	8	
Complex Karyotype	Present	26	63.4	15	36.6	3	0.3
	Absent	1	33.3	2	66.7	38	

AML-CI: acute myeloid leukemia comorbidity index; n: number of patients; ORR: overall response rate. Cytogenetics evaluable for 41 patients.

**Table 2 jcm-14-05007-t002:** Cox analysis regarding DFS.

Variable	HR	95% CI (Lower–Upper)	*p*-Value	HR	95% CI (Lower–Upper)	*p*-Value
Gender (F vs. M)	1.904	0.426–8.519	0.400			
Age	1.076	0.375–3.088	0.891			
AML-CI Risk (high vs. others)	1.076	0.375–3.088	0.891			
Cytogenetic risk	0.06	0.0001–65	0.4			
Complex karyotype	1.5	0.3–6.4	0.6			
**Treatment (DEC vs. AZA)**	**11.304**	**1.335–95.749**	**0.026**	**16.372**	**1.295–207.046**	**0.031**
*NPM1* mutation (yes vs. no)	0.041	0.000–6101.159	0.599			
**WBC (per 10,000 unit increase)**	**1.87**	**1.2–11.5**	**0.015**	1.4	0.9–11.2	0.288
Platelets (per 10,000 unit increase)	0.9	0.6– 1.1	0.356			
Hemoglobin (per 0.1 unit increase)	0.095	0.0484–1.0867	0.884			
Therapy duration	0.871	0.697–1.089	0.226			
Washout duration	0.980	0.889–1.081	0.693			
VIALE-A trial exclusion (yes)	1.859	0.414–8.357	0.419			

AML-CI: acute myeloid leukemia comorbidity index; DEC: decitabine; AZA: azacitidine; WBC: white blood cells; 95% CI: 95% confidence intervals.

**Table 3 jcm-14-05007-t003:** Median Overall Survival by Clinical and Biological Parameters.

Parameter	Median OS (months)	95% CI (Months)	*p*-Value
Overall cohort	7.7	5.9–9.4	
**Exclusion from VIALE-A trial**			**0.021**
**No**	**8.4**	**–**	
**Yes**	**5.8**	**1.4–10.2**	
Gender			0.691
Female	7.7	5.8–9.7	
Male	7.7	3.4–12.1	
Antifungal prophylaxis			0.576
No	8.3	7.3–9.3	
Yes	6.6	6.2–12.5	
Treatment (AZA vs. DEC)			0.948
AZA	7.7	2.5–12.7	
DEC	7.6	5.5–13.1	
*NPM1* mutation			0.439
Absent	7.6	5.6–9.6	
Present	9.7	–	
Performance Status (PS 0–2 vs. 3–4)			0.069
PS 0–2	8.4	7.4–9.4	
PS 3–4	5.8	1.8–9.8	

OS: overall survival; PS: performance status; AZA: azacitidine; DEC: decitabine; 95% CI: 95% confidence intervals.

**Table 4 jcm-14-05007-t004:** Cox Regression Analysis of Overall Survival.

Parameter	HR	95% CI	*p*-Value
Gender (M vs. F)	0.85	0.39–1.87	0.692
AML CI risk	1.04	0.61–1.79	0.889
HB (per unit)	0.82	0.60–1.12	0.207
PLT (per unit)	1.00	1.00–1.00	0.320
WBC (per unit)	1.00	1.00–1.00	0.966
**Exclusion VIALE-A trial**	**2.50**	**1.12–5.61**	**0.026**
Treatment (DEC vs. AZA)	0.97	0.44–2.18	0.948
Antifungal prophylaxis	1.24	0.58–2.65	0.578
Cytogenetic risk	1.17	0.6–2.3	0.6
Complex karyotype	0.6	0.35–6.37	0.6
*NPM1* mutation	1.00	0.37–2.72	0.997
Washout duration	0.98	0.92–1.05	0.611
Treatment duration	0.91	0.77–1.08	0.276

AZA: azacitidine; DEC: decitabine; 95% CI: 95% confidence intervals; HB: Hemoglobin; WBC: white blood cells; PLT: platelets.

## Data Availability

The raw data supporting the conclusion of this article will be made available by the authors without undue reservation.

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
