# Peer review of "Length of Washout Period After Remission Does Not Influence Relapse Risk in Patients with Acute Myeloid Leukemia Treated with Hypomethylating Agents Combined with Venetoclax"

_jcm, 2025, doi:10.3390/jcm14145007_

Round 1

Reviewer 1 Report

Comments and Suggestions for Authors

The study design is appropriate ,yet I think the patient's group is a little bit small .

As you mentioned the study lacks  MRD assessment ,that to assure complete remission.

Really, references are very updated which adds to the work. 

Reviewer 2 Report

Comments and Suggestions for Authors

Journal: Journal of Clinical Medicine
Manuscript ID: JCM-3708200
Title: "Length of washout period after remission does not influence relapse risk in patients treated with hypomethylating agents combined with venetoclax"

  • Inconsistent use of "VIALE-A" (sometimes with "trial", sometimes without). Standardize as VIALE-A trial throughout.
  • Alternation between "HMA-VEN" and "VEN-HMA". Use VEN-HMA (alphabetical order) for consistency.
  • Page 5: "median platelets were 34000/uL, ranged between 2000/uL and 13000/uL" : 34,000/uL with a range of 2,000–13,000/uL? Correct values or clarify discrepancy.

  • Table 1: Percentage totals for "AML-CI" do not align with *n=44* (15.9 + 27.3 + 25 + 31.8 = 100%, but absolute values must match cohort size).

  • "More date coming from real-world analysis" → Correct to "More data from real-world analyses".

  • Figure 1: Legend lists *N=21* (eligible) and *N=23* (ineligible), but total should be 44. Clarify if patients were excluded from analysis.

  • 21 patients received azole antifungals (CYP3A4 inhibitors). Did this influence washout duration or toxicity? Was venetoclax dose adjustment required?
  • "With only 44 patients and 7 relapses in responders (n=27), was your study powered to detect washout-associated differences in DFS/OS?"
  • You report 'median platelets: 34,000/μL (range: 2,000–13,000/μL)'. Is this range correct? If so, how do you interpret a median outside the reported interval?
  • The mean washout was 20±25 days (extremely wide range). How did you clinically manage patients with exceptionally long washouts (>40 days)? Was any duration threshold associated with increased relapse risk?

Reviewer 3 Report

Comments and Suggestions for Authors

In the current analysis, the authors investigated the clinical findings of patients with acute myeloid leukemia (AML) who received venetoclax (VEN) and hypomethylating agents (HMA). The results are interesting and valuable for readers in this field. However, there are several issues in the manuscript that need to be addressed.

  1. In Table 3, the authors state that the best response to therapy was associated with the clinical course. However, this response could not have been available at the first day of treatment, which is defined as the starting point for overall survival (OS) in the current study. Therefore, the authors should consider alternative statistical approaches for this analysis.
  2. In Table 1, the authors examined the relationships between remission rates and several clinical variables, and found that eligibility for the VIALE-A trial and performance status (PS) were associated with higher remission rates. However, PS is a component of the VIALE-A eligibility criteria, as described in the manuscript, and therefore represents a confounding factor in this analysis.

Minor issues:

  1. (Title) Including “AML” in the title would help readers clearly understand the focus of the manuscript.
  2. (Line 92) Please verify whether the term “prospective” is appropriate in this context.
  3. (Lines 96–103) A figure illustrating the therapeutic strategy could help readers better follow the treatment approach.

Round 2

Reviewer 3 Report

Comments and Suggestions for Authors

The authors have revised the manuscript based on the reviewers’ comments, and the changes have improved the clarity of the manuscript.

One minor point is that the authors did not include “AML” in the title, although they appeared to agree with the reviewer’s suggestion.

Author Response

Thank you for your appreciation. We modified the title according to your suggestion